# Generation of Stable Epithelial–Mesenchymal Hybrid Cancer Cells with Tumorigenic Potential

**DOI:** 10.3390/cancers15030684

**Published:** 2023-01-22

**Authors:** Roslyn Tedja, Ayesha B. Alvero, Alexandra Fox, Carlos Cardenas, Mary Pitruzzello, Hussein Chehade, Tejeshwhar Bawa, Nicholas Adzibolosu, Radhika Gogoi, Gil Mor

**Affiliations:** 1C.S. Mott Center for Human Growth and Development, Department of Obstetrics and Gynecology, Wayne State University, Detroit, MI 48201, USA; 2Department of Obstetrics and Gynecology, Family HealthCare Network, Porterville, CA 93257, USA; 3Department of Dermatology, Yale Medical School, New Haven, CT 06510, USA; 4Center for Molecular Medicine and Genetics, Wayne State University, Detroit, MI 48201, USA; 5Department of Physiology, Wayne State University, Detroit, MI 48201, USA; 6Karmanos Cancer Institute, Wayne State University, Detroit, MI 48201, USA

**Keywords:** EMT, epithelial, mesenchymal, ovarian cancer, E/M hybrid

## Abstract

**Simple Summary:**

Cancer spreading into different organs, or cancer metastasis, remains the major clinical problem in almost all cancer types. Cancer metastasis is known to be governed by cancer cell adaptation and differentiation in the process known as epithelial-to-mesenchymal transition (EMT). Recently, this process has been shown to yield to epithelial and mesenchymal like properties, resulting in what is known as epithelial/mesenchymal (E/M) hybrid cancer cells. However, the characteristics of E/M hybrid cells, their importance in tumor progression, and the key regulators in the tumor microenvironment that support this phenotype are still poorly understood, especially in ovarian cancer. In this study, we aim to dissect and characterize the different steps during the transition of cancer cells into the E/M hybrid state. Our data show that once the cells enter the E/M hybrid state, they acquire stable anoikis resistance, invasive capacity, and tumorigenic potential. We identified the hepatocyte growth factor (HGF)/c-MET pathway as a major driver that pushes cells in the E/M hybrid state.

**Abstract:**

Purpose: Cancer progression, invasiveness, and metastatic potential have been associated with the activation of the cellular development program known as epithelial-to-mesenchymal transition (EMT). This process is known to yield not only mesenchymal cells, but instead an array of cells with different degrees of epithelial and mesenchymal phenotypes with high plasticity, usually referred to as E/M hybrid cells. The characteristics of E/M hybrid cells, their importance in tumor progression, and the key regulators in the tumor microenvironment that support this phenotype are still poorly understood. Methods: In this study, we established an in vitro model of EMT and characterized the different stages of differentiation, allowing us to identify the main genomic signature associated with the E/M hybrid state. Results: We report that once the cells enter the E/M hybrid state, they acquire stable anoikis resistance, invasive capacity, and tumorigenic potential. We identified the hepatocyte growth factor (HGF)/c-MET pathway as a major driver that pushes cells in the E/M hybrid state. Conclusions: Herein, we provide a detailed characterization of the signaling pathway(s) promoting and the genes associated with the E/M hybrid state.

## 1. Introduction

Cancer progression and metastasis are known to be controlled by the epithelial-to-mesenchymal transition (EMT) process [1]. EMT is a step-wise process, which commences with (1) the loss of apical-basal cell polarity; (2) disruption of cell-to-cell interaction; (3) extracellular matrix (ECM) disassembly, which results in degradation of the basal membrane; (4) ECM reorganization, followed by (5) reorganization of the actin cytoskeleton [2,3,4,5]. This complex process is driven by an orchestra of transcription factors including genes in the SNAIL, TWIST, and Zeb families. 

EMT is not a simple one-directional process wherein epithelial (E) cells become mesenchymal (M) cells. Instead, it is a bi-directional dynamic process, which along the way yields cells in various degrees of the E/M spectrum and with a high degree of plasticity. Cells in this state have been referred to as being in an E/M hybrid state [6]. The characteristics of these E/M hybrid cells, their importance in tumor progression, and the key regulators in the tumor microenvironment that support this phenotype are still poorly understood.

Ovarian cancer, the most lethal of the gynecological cancers, represents a unique form of metastatic process that differs with other solid tumors [7,8]. The metastasis route of ovarian cancer is distinctly different to that of other epithelial cancers, in which it metastasizes as the cells are exfoliated from the primary tumor directly into the peritoneal cavity, (transcoelomic metastasis) and attaches to the mesothelium and the adipocyte tissue within the peritoneum. Consequently, the utilization of classical solid tumor models of EMT (such as breast cancer) to comprehend the ovarian cancer metastatic process has failed to advance our understanding of the complexity of intra-peritoneal carcinomatosis [9]. In addition, the existing in vitro models of ovarian cancer cells do not fully recapitulate the full spectrum of the EMT process.

We developed in vitro and in vivo models of EMT and carcinomatosis to further understand the early stages in epithelial–mesenchymal transformation in ovarian cancer. In this study, we performed transcriptome analysis (RNA whole genome sequencing) on ovarian cancer cells in the different stages of EMT and characterized their unique properties related to anoikis resistance, cellular remodeling, and tumorigenic potential, and identified the signal(s)/mechanism(s) that give rise to the E/M hybrid state. We demonstrate that hepatocyte growth factor (HGF) is a major driver that pushes cells in the E/M hybrid state. Furthermore, we report that the E/M hybrid state possess anoikis resistance, invasive capacity, and tumorigenic potential. 

## 2. Materials and Methods

### 2.1. Cell Lines and Culture Conditions

Sample collection was performed with patient consent and approved by the Human Investigation Committee of Yale University School of Medicine. Ovarian cancer cells were isolated from patients diagnosed with Stage III/IV serous ovarian carcinoma. These in-house derived ovarian cancer cells (R182, R2615, 01-28, and Tara R182) have been previously described [10,11,12,13,14,15]. Ovarian cancer cell lines (OVCAR3, OVCA432, SKOV3, and A2780) were purchased from ATCC. Cells were cultured in RPMI media supplemented in 10% FBS and grown at 37 °C with 5% CO_2_. All of the cell lines tested negative for mycoplasma, authenticated once a year by STR profiling, and used within six passages between experiments.

### 2.2. Antibodies and Reagents

The antibodies used are listed in Appendix A. The recombinant human hepatocyte growth factor (rHGF) was purchased from Peprotech, Inc; Cranbury, NJ, USA.

### 2.3. Anoikis Assay

The anoikis and cell viability assays were performed as described in our previous publication [16]. To confirm cell viability following the anoikis assay, the cells were cultured back to the tissue culture-treated plate. These cells were designated as HGF-R182. 

### 2.4. Invasion Assay

Invasion assays were performed in 96-well plates with growth-factor reduced Matrigel (Corning). A total of 103 cells were cultured in 50% Matrigel/RPMI1640 media in a humidified (37 °C, 5% CO_2_) BioTek BioSpa live cells analysis system (BioTek, Santa Clara, CA, USA). The cells were monitored and imaged every 4 h for a total of 72 h using the Cytation 5 cell imaging system (BioTek, Santa Clara, CA, USA). 

### 2.5. RNA Sequencing

The total RNA samples prepared using the TRIZOL reagent were used as the input. The RNA-Seq library was enriched using the KAPA mRNA HyperPrep Kit (KAPA Biosystem, Wilmington, MA, USA). The multiplex library was then subjected to high-throughput sequencing using HiSeq 4000 platforms (Illumina; San Diego, CA, USA). Differential expression analysis between the primary epithelial group with all other groups was conducted using the limma R package. Raw counts were normalized using the “limma-trend” approach. Differentially expressed genes were identified by treating the epithelial group as a fixed effect variable and the subjects as a random effect variable. The data were further analyzed using iPathway software. The GO enrichment analyses were plotted using the R package, ggplot2. 

### 2.6. q-PCR

Total RNA was extracted using RNeasy Mini Kits (Qiagen, Austin, TX, USA) according to the manufacturer’s instructions. cDNA was synthesized with the iScript cDNA Kit (Bio-Rad, Hercules, CA, USA). Quantitative PCR was performed using SYBR Green Supermix (Bio-Rad, Hercules, CA, USA), followed by detection with the CFX96TM PCR detection system (Bio-Rad, Hercules, CA, USA). GAPDH was used as the reference gene. Relative expression was calculated using the comparative ΔΔCT method. All reactions were performed in triplicate. Primer sequences are included in Appendix A.

### 2.7. Protein Extraction and Quantification

Protein extraction and quantification using the BCA assay were carried out as previously described [11,17]. 

### 2.8. SDS Polyacrylamide Gel Electrophoresis and Western Blot Analysis

Twenty μg of the protein samples was denatured in the sample buffer and electrophorized on a 12% SDS–polyacrylamide gel as previously described. The blots were imaged using GE ImageQuant LAS 500 chemiluminescence (Cytiva Life Sciences, Marlborough, MA, USA).

### 2.9. Tumorigenic Potential In Vivo Assay

All of the in vivo studies described here have been approved by Wayne State University Animal Care and Use Committee. To assess the tumorigenic potential, four million ovarian cancer cells (epithelial cells: OVCAR3, R182, R2615; E/M hybrid cells: OVCAR3, R182, R2615, HGFR182; mesenchymal cells: OVCAR3, R182, R2615, OCSC1-F2) were injected intraperitoneally (i.p.) to athymic nude-Foxn1nu mice. All animals were monitored weekly for their body weight and palpated for signs of tumor growth. Animals were sacrificed 66 days after cell inoculation (for epithelial cells: R182, R2615, OVCAR3) or 33 days after cell inoculation (for E/M hybrid cells: R182, R2615, OVCAR3, HGFR182; mesenchymal cells: OVCAR3, R182, R2615, OCSC1-F2). 

### 2.10. Statistical Analysis

The unpaired two-tailed Student *t*-test, one-way analysis of variance (ANOVA) with Dunnett’s multiple comparison or two-way ANOVA with Sidak multiple comparison were used to compare the different groups. Data of each group were collected with at least three biological repeats. The variances in each group were calculated to make sure that they were similar between the compared groups. *p* values of 0.05 or less were considered statistically significant. All of the statistical analysis was performed using Prism 9 software (GraphPad Software, San Diego, CA, USA). The illustrations and scheme were produced using Biorender software. 

## 3. Results

### 3.1. Characterization of Epithelial Ovarian Cancer Cell Lines Based on Their Epithelial and Mesenchymal Markers

The process of EMT has been extensively studied in ovarian cancer using different cell lines with the assumption that these cells are in the same epithelial state. We evaluated a panel of widely used epithelial ovarian cancer cell (EOCC) lines for the expression of epithelial or mesenchymal protein markers. The EOCC lines used included the in house isolated clones (clones: R182, OCSC1-F2, 01-28, R2615) [15,18] and ATCC obtained cell lines (OVCAR3, OVCA432, SKOV3, A2780). For the epithelial phenotype, we used keratin-18 (Krt18), β-catenin, E-cadherin, and Claudin-3; for the mesenchymal phenotype, we included N-cadherin, Vimentin, and Twist1. 

Interestingly, we observed that only four cell lines (R182, R2615, OVCAR3, and OVCA432) were positive for Krt18 and β-catenin protein expression. OVCAR3 and OVCA432 were Krt18 and β-catenin negative, but E-cadherin and Claudin-3 positive (Figure 1A). On the other hand, OCSC1-F2, 01-28, and A2780 were negative for all of the epithelial markers tested, but were positive for mesenchymal markers Vimentin and Twist1 (Figure 1A). Therefore, although they are considered as EOCCs, this is not the case and the different cell lines may represent different stages of cell differentiation. To further characterize the phenotype of these epithelial ovarian cancer cells, we classified them into three main groups: (1) Krt18+, β-catenin+, and N-cadherin+ (R182 and R2615); (2) Krt18+, β-catenin+, E-cadherin+, Claudin-3+, and Vimentin+ (OVCAR3 and OVCA432); and (3) Krt18-, β-catenin-, E-cadherin-, Claudin-3-, Vimentin+, and Twist+ (SKOV3, F2-R182, 01-28, A2780). Each of these cell lines were tested for their capacity to resist anoikis, invade, and their tumorigenic potential. 

To test for anoikis resistance, we seeded the cells in ultra-low adherent (ULA) plates and monitored the survival and spheroid formation for a period of 7 days (Figure 1B). The cells in group 1 (R182, R2615) were sensitive to anoikis, as revealed by the absence of viable cells (Figure 1B, Appendix A). Cells in groups 2 and 3 were resistant to anoikis, as shown by the presence of viable cells and the formation of cell aggregates and spheroids (Figure 1B, Appendix A). 

To test the invasion potential, cells were seeded within 50% Matrigel and the invasion capacity was monitored by live imaging. Groups 1 and 2 did not demonstrate the capacity to invade (Figure 1B, Appendix A) and only group 3 showed a high degree of invasion capacity characterized by the presence of projections and the formation of spindle-like structures (Figure 1B, Appendix A).

We also determined the tumorigenic potential by injecting 5 × 10^6^ cells intraperitoneally (i.p.) in balb/c nude mice. Tumor formation was determined 33 days post injection. Only group 3 showed tumorigenic capacity and formed extensive carcinomatosis within the peritoneal cavity (Figure 1B). No tumors were detected in the animals injected with cells from groups 1 and 2 (Figure 1B). Thus, in addition to the differences in the expressed E/M markers, these three groups also possess different pro-tumor properties. Taken together, we can therefore define at least three different ovarian cancer states. Group 1 represents the epithelial phenotype with anoikis sensitivity and the absence of both invasive potential and tumorigenic potential. Group 2 represents the E/M hybrid state, which co-expresses epithelial and mesenchymal markers, possesses resistance to anoikis, but still lacks invasive capacity and tumorigenic potential. Finally, group 3 represents the mesenchymal phenotype, characterized by the expression of mesenchymal markers, anoikis resistance, invasiveness, and tumorigenic potential. 

### 3.2. Identification of a Tumorigenic E/M Hybrid State in Ovarian Cancer

Since EMT does not result in a single mesenchymal state, but rather in a multiplicity of hybrid states with several degrees of epithelial and mesenchymal features, our next objective was to characterize the early molecular and cellular changes taking place in the process of EMT that lead to the generation of cancer cells with metastatic and tumorigenic potential. In order to dissect the pathways responsible for early changes, we used a previously reported model of EMT consisting of EOCCs (group 1 R182 and R2615, anoikis sensitive, non-invasive, and non-tumorigenic) cultured in high confluence in low-serum conditions for 12 days [11,19,20,21]. Under these conditions, anoikis sensitive EOCCs undergo morphological changes characteristic of EMT. After 48 h, we observed a change in the cell morphology characterized by the acquisition of fibroblastic shape. By day 12, we observed the formation of islands, and the presence of detached viable spheroids (Figure 2A) [11]. 

Thus, in this model, we can define three states of differentiation based on cellular morphology: (1) epithelial state; (2) E/M hybrid state (at 48 h); and (3) spheroid or mesenchymal state (9–12 days) (Figure 2A). We then defined the molecular changes on these different states by evaluating the expression of epithelial (β-catenin, KRT18, Epcam) and mesenchymal (Sox2, Twist1, Vimentin, Slug) markers at both the mRNA and protein level. Compared to the epithelial cells, the cells in the E/M hybrid state expressed lower levels of KRT18, higher levels of Epcam, and similar levels of β-catenin (Figure 2B,C). In addition, the E/M hybrid cells remained negative for Twist1, but gained Sox2, Slug, and Vimentin, which were not present in the epithelial cells (Figure 2B,C). Characterization of the cells in the spheroids showed gain in Twist1, Vimentin, and downregulation of all epithelial markers tested. Interestingly, the mesenchymal cells showed lower levels of Sox2 and Slug compared to the E/M hybrid cells (Figure 2B), demonstrating that different stages of EMT are associated with varying E/M markers. 

Having established the molecular characteristics of these three states, we evaluated their functionality by determining anoikis resistance, invasion, and tumorigenic capacity. Whereas epithelial cells did not survive the detached conditions, the E/M hybrid cells demonstrated anoikis resistance, as evidenced by the presence of viable spheroids in the ULA wells (Figure 2D). Compared to epithelial cells, the E/M hybrid cells showed a more invasive phenotype marked by the formation of projections throughout the matrix (Figure 2E). The invasive capacity was quantified by measuring cell confluence at a 72 h time point and our results showed that there was a four-fold increase in confluence in the E/M hybrid compared to the epithelial cells (Figure 2F). 

We differentiated three EOCC lines (R182, R2615, OVCAR3), and the derived E/M hybrid and mesenchymal cells from each line were all tested. As shown in Figure 2G, the epithelial cells did not form tumors, while their respective E/M hybrid and mesenchymal cells demonstrated tumorigenic potential by the formation of multiple i.p. tumor implants. 

These results show that although the E/M hybrid cells conserved some of their epithelial phenotypes, these cells underwent a process of differentiation that is committed to mesenchymal transformation, and under specific conditions, developed anoikis resistance and tumorigenic potential. Since these cells (1) exhibited an epithelial morphology; (2) maintained the expression of classical epithelial markers (i.e. β-catenin); (3) had not fully acquired classical mesenchymal markers such as Twist1 and Slug; and (4) had acquired anoikis resistance and tumorigenic potential, we reasoned that these cells may indeed represent the E/M hybrid state [22]. Based on these observations, we postulated a two-phase model of EMT in EOCCs (Figure 2A) wherein Phase I is the acquisition of an E/M hybrid state characterized by an epithelial morphology but with tumorigenic potential, and Phase II is the acquisition of a fully mesenchymal phenotype characterized by mesenchymal morphology, loss of Ck18, reduction in β-catenin, gain in Twist1, anoikis resistance, and tumorigenic potential. 

### 3.3. Characterization of E/M Hybrid State 

To fully characterize the molecular signature of cells in the E/M hybrid state, we performed bulk RNA sequencing and compared the transcriptome between the three EOCCs (OVCAR3, R182, R2615) (CK18+/β-catenin+/Slug−/Twist1−) with their corresponding E/M hybrid and mesenchymal cells. We first focused our attention on characterizing the molecular phenotype of the E/M hybrid state by comparing CK18+/β-catenin+/Slug−/Twist1− EOCCs with CK18+/β-catenin+/Slug+/Twist1− E/M hybrid cancer cells (48 h differentiated). Remarkably, we observed major differences between the two stages; there were 209 differentially expressed genes (DEGs) (Figure 3B) and 51 differentially expressed pathways (Figure 3C). From these pathways, there were four differentially expressed pathways related to EMT including TGF-β, hippo signaling, focal adhesion, and axon guidance pathways (Figure 3D). 

Further GO enrichment analysis for biological processes was performed; there were 384 processes that were differentially regulated. Twenty of these differentially expressed biological processes are related to EMT including ECM disassembly and organization, positive regulation of NIK/NF-κB signaling, cellular response to TGF-β stimulus, p38 MAPK cascade, and activation of protein kinase B (Figure 3E). In addition, GO enrichment analysis for cellular components and molecular functions also showed that ECM, basement membrane, collagen, and integrin binding were differentially regulated in the E/M hybrid compared to the epithelial cells (Appendix A). Therefore, we can conclude that the cells in the E/M hybrid state have undergone: (1) ECM remodeling (including disassembly and collagen fibril organization); (2) actin-cytoskeleton remodeling that may prepare the cells for detachment from the basement membrane; (3) negative regulation of the extrinsic apoptotic pathway in the absence of the ligand; and (4) the reorganization of ECM (Figure 3E).

### 3.4. Characterization of Late EMT Differentiation Stage 

We then compared the differential gene expression between the three CK8+/β-catenin+/ Slug−/Twist1− EOCCs with their derived mesenchymal state (sOVCAR3, sR182, sR2615): Phase II (Figure 1A). We observed a total of 3860 DEGs (Figure 4B). Pathway analysis revealed 57 differentially regulated pathways (Figure 4C) and 11 of them were involved in both canonical (regulation of actin cytoskeleton, focal adhesion, ECM-interaction, tight and gap junctions) and non-canonical (pyruvate metabolism, glycolysis/gluconeogenesis, VEGF signaling) EMT signaling (Figure 4D).

The transition from an epithelial to a mesenchymal state is further supported by data from the GO enrichment analysis for biological processes. Comparing the epithelial to mesenchymal states, there were 335 differentially regulated biological processes. The top 20 differentially regulated processes were likewise connected to EMT. These biological processes include actin cytoskeleton reorganization, integrin activation, epithelial cell–cell adhesion, cell–matrix adhesion as well as the positive regulation of EMT (Figure 4E). Further GO enrichment analysis for cellular components and molecular functions highlight the significant changes in plasma membrane, adherent junctions, protein kinase binding, and Rac GTPase binding (Appendix A).

Interestingly, while the TGF-β signaling pathway was differentially regulated in the E/M hybrid compared to epithelial cells, this pathway was not differentially regulated when the epithelial cells were compared to spheroids (Figure 4D), suggesting a role for TGF-β signaling only in the early stage of differentiation. Thus, together with the Western blot results shown in Figure 2C, these findings strongly demonstrate that the spheroids represent a mesenchymal state, characterized by anoikis resistance and EMC reorganization (Figure 4E).

### 3.5. ECM Disassembly and Organization Are the First Step in Early Phase of EMT

As above-mentioned, the first step in Phase I of EMT is ECM organization and disassembly. Therefore, we sought to determine the DEGs responsible for the ECM disassembly process in the E/M hybrid state. Compared to its epithelial origin, E/M hybrid cells showed increased expression of Matrix Metallopeptidases (MMPs), MMP3 and MMP1, as detected by the transcriptome analysis (Figure 5A) and validated by q-PCR (Figure 5B). Furthermore, based on the transcriptome analysis, the genes associated with basal membrane and ECM were significantly downregulated in the E/M hybrid cells such as collagens including COL6A1, COL4A1, and COL4A2 (Figure 5C), while the expression of Tenascin-C (TNC) was significantly increased in the E/M hybrid state cells compared to their epithelial counterparts (Figure 5C). Significantly higher levels of TNC mRNA and protein were observed in the E/M hybrid cells compared to the epithelial cells (Figure 5D,E). Quantification of intra-cellular and secreted TNC showed significantly higher intra-cellular TNC in the E/M hybrid cells (380 ± 13 pg TNC/mg total cell lysate) compared to the epithelial cells (105 ± 1 pg TNC/mg cell lysate) (Figure 5F). Secreted TNC was likewise higher in the E/M hybrid cells (4890 ± 165 pg TNC/106 cells vs. 850 ± 38 pg/106 cells in epithelial cells) (Figure 5G). Taken together, these data indicate that some important characteristics of the E/M hybrid state are the expression of metalloproteases, downregulation of collagen expression, and upregulation of intracellular and secreted TNC. Together, these changes may facilitate cell dissociation from the ECM and will provide an early preparation for the cells to respond to further environmental signals responsible for the cascade of events leading to the generation of mesenchymal cells.

### 3.6. Rearrangement of Focal Adhesion Is the Next Step in E/M Hybrid State

Focal adhesions are protein structures that form a link between the actin cytoskeleton and the ECM, which allow for communication between cells and its extracellular environment. The process of ECM reorganization, detachment, and migration requires the presence of specific signaling pathways that will modify the genes maintaining the characteristics of the epithelial cells. Based on the biological pathway analysis from DEGs, focal adhesion reorganization is one of the significant biological pathways as the epithelial cells enter the E/M hybrid state (Phase I) (Figure 3D). We identified three important genes related to focal adhesion reorganization that were differentially expressed in the E/M hybrid state compared to their epithelial origin: HGF, TNC, and VAV3 (Figure 6A). HGF mRNA expression was increased by 22- and 12-fold in R182 and R2615, respectively, as they enter the E/M hybrid state compared to the epithelial state confirmed by qPCR (Figure 6B). While there was no detectable HGF at the protein level in epithelial cells, we observed a significant increase in HGF secretion in the E/M hybrid cells (85 ± 4.6 and 189 ± 12 pg/10^6^ cells at 24 and 48 h, respectively) (Figure 6C). HGF is known to bind to its receptor, c-MET, which is highly expressed in EOCCs (CQ value of 15) and its expression was not affected by the differentiation process (Figure 6B). These findings suggest that HGF signaling is activated in the early stages of differentiation and that the E/M hybrid cells may sustain themselves with continuous HGF secretion. Two additional genes associated with the rearrangement of focal addition were differentially expressed, TNC and VAV3 (Figure 6A). Validation of these two genes revealed that mRNA expression increased in the E/M hybrid cells detected by q-PCR (Figure 5D and Figure 6B). 

We sought to determine the signaling pathway(s) that facilitate the differentiation process, which yield cells in the E/M hybrid state with anoikis resistance and the ability to invade and form tumors. Based on the GO-enrichment analysis comparing E/M hybrid cells and epithelial cells, the most significant biological process is the regulation of signaling receptor activity (Figure 3E). The genes in this process and their log fold-change are listed in Figure 6D. The most differentially expressed gene was HGF, therefore, we pursued this further to investigate the role of HGF in the differentiation process.

### 3.7. HGF Induce Epithelial Cells to Enter the E/M Hybrid State 

In light of the finding of HGF upregulation in several of the critical pathways related to the early stages of EMT, we then aimed to further investigate whether HGF had a role in the formation of cells in the E/M hybrid state. Specifically, we determined whether HGF can induce ECM reorganization, focal adhesion rearrangement, and anoikis resistance. First, we tested whether the HGF/MET pathway is functional in EOCCs by determining whether treatment with HGF is able to induce the phosphorylation of its receptor, c-MET, and/or the components of the MAP/ERK signaling pathway. Thus, EOCCs (R182) were treated with 80 ng/mL human recombinant HGF (rHGF) at different timepoints. As shown in Figure 7B, rHGF induced c-Met phosphorylation beginning at 15 min post-treatment and persisted until 60 min. Phosphorylated c-MET was not detected in the non-treated control (Figure 7B). HGF did not have an effect on the levels of the total c-MET (Figure 7B). Furthermore, the phosphorylation of ERK following rHGF treatment beginning at 15 min post-treatment was maintained up to 120 mins, which was the last timepoint (Figure 7B).

After we confirmed that the HGF/c-MET pathway is functional in these cells, we then proceeded to investigate whether rHGF treatment could induce the process of EMT (Figure 7C). R182 (CK18+/β-catenin+) cells were treated with 80 ng/mL rHGF for 48 h. We then evaluated the expression of epithelial and mesenchymal markers. rHGF treatment did not significantly affect the expression of epithelial markers CK18 and β-catenin; however, it did induce the expression of SOX2, Slug, and Twist1, known regulators of mesenchymal transformation [17,23,24]. Based on the changes observed in the epithelial and mesenchymal markers, we can conclude that rHGF promotes the differentiation of epithelial cancer cells into the E/M hybrid state.

### 3.8. HGF Induces Anoikis Resistance in EOCCs

We then determined whether cells treated with rHGF had acquired the functional properties observed in the E/M hybrid cells such as anoikis resistance, invasive potential, and tumorigenicity. We first evaluated the effect of rHGF treatment on anoikis resistance. Epithelial cells, R182, were cultured in ultra-low attachment plates in the presence or absence of 80 ng/mL rHGF. The presence of rHGF induced anoikis resistance in the epithelial cells and led to the formation of viable spheroids (Figure 7D). To confirm that the observed resistance to anoikis was due to the effect of HGF, we treated the cells with rHGF (80 ng/mL) in the presence or absence of the c-MET inhibitor, Crizotinib (100 nM). As shown in Figure 7D(iii), the presence of Crizotinib abolished the pro-survival effect of HGF. These findings clearly support the premise that the HGF/c-Met axis plays a role in conferring anoikis resistance in EOCCs, a critical stage in the establishment of the E/M hybrid state. 

### 3.9. Role of HGF on ECM Assembly and Focal Adhesion

As shown in Figure 3D,E, another important pathway associated with the E/M hybrid state is ECM reorganization and focal adhesion. We next investigated whether HGF could induce these changes by determining the expression of MMPs including MMP-1 and -3 as well as TNC, which we have shown to be upregulated in the E/M hybrid cells. Thus, EOCCs (clones R182 and R2615) were treated with rHGF (80 ng/mL) for 48 h and mRNA levels of MMP-1, -3, TNC, HGF, and c-MET were determined by qPCR. As shown in Figure 7H, rHGF induced the mRNA expression of MMP1 MMP3, and TNC, but not HGF or c-MET. The upregulation in TNC was confirmed at the protein level. The levels of secreted TNC were significantly increased in the epithelial cells treated with 80 ng/mL rHGF for 48 h (3800 ± 46 pg/106 cells vs. 1000 ± 37 pg/106; rHGF and the control, respectively) (Figure 7I). These data demonstrate that HGF is sufficient to induce ECM reorganization and focal adhesion. 

### 3.10. HGF Promotes Matrix Degradation and Invasion Capacity in EOCCs

Having shown that HGF induced the expression of MMPs, TNC as well as genes associated with ECM reorganization, we tested whether HGF could promote invasion potential. To test this hypothesis, epithelial cells (clone R182 and R2615) were seeded in the presence or absence of 80 ng/mL rHGF. In the absence of rHGF, the epithelial cells did not show the ability to invade and degrade the matrix. However, in the presence of HGF, the epithelial R182 and R2615 clones revealed a significant increase in the invasion capacity, as shown by the formation of projections within the Matrigel (Figure 7F, Appendix A). In addition to these two EOCCs, we also tested the role of HGF on OVCA432, which has a limited capacity to invade (Appendix A), by exposing the cells to rHGF as well as OVCA433, which has the capacity to invade (Appendix A), and by exposing the cells to either rHGF or Crizotinib (100 nM). Interestingly, the addition of rHGF in OVCA432 further enhanced the invasion capacity while the presence of the c-MET inhibitor (Crizotinib) in OVCA433 significantly inhibited the invasion process (Appendix A). These findings highlight the importance of HGF signaling in ECM reorganization and invasiveness in ovarian cancer cell lines.

### 3.11. HGF Induce Reprograming in the Epithelial Ovarian Cancer Cells

Our next objective was to determine whether the changes observed with HGF were transient or would establish a persistent phenotype. Thus, epithelial cells were cultured in the presence oof rHGF (Figure 8A(i)) first in attached conditions for 48 h, followed by culturing in ULA as spheroids for 7 days (Figure 8A(ii)). Afterward, the rHGF was removed and the spheroids were allowed to re-attach and grow as a monolayer (referred to here as HGFR182) (Figure 8A(iii)). We then tested the anoikis resistance of HGFR182 by transferring the cells to ULA and culturing in the absence of rHGF. As shown in Figure 8A(iv), HGFR182 acquired the capacity to survive in low attachment despite the absence of rHGF in the media. These results demonstrate that HGF treatment induces a persistent change that can confer anoikis resistance. Additionally, the invasive capacity of HGFR182 was also tested in the absence of rHGF. As shown in Figure 8B, the HGFR182 cells were able to invade (Figure 8C). These data indicate that the HGFR182 cells did not only acquire persistent anoikis resistance, but also had invasive capacities.

### 3.12. HGF-Induced E/M Hybrid State Cells Acquire Tumorigenic Potential

Having demonstrated that HGF promotes anoikis resistance and invasiveness, we tested whether these characteristics were sufficient to confer tumorigenic potential. Hence, we injected epithelial OC cells and HGF-induced E/M hybrid state cells (HGFR182) i.p. in balb/c nude mice and monitored the tumor formation. In contrast to EOCCs, which did not form tumors, HGFR182 demonstrated a 100% tumor formation rate. All eight mice injected with HGFR182 formed tumors with six of them forming carcinomatosis with more than 15 detectable implants, one mouse with 14 implants, and the remaining mouse with two implants (Figure 8E). These results clearly show that the HGF-induced E/M hybrid state does not only confer acquire anoikis resistance and invasiveness, but more importantly, tumorigenic potential. 

### 3.13. Characteristics of the HGF-Induced E/M Hybrid State EOCCs

Finally, we sought to determine the characteristics of the HGF-induced E/M hybrid state. To achieve this, we performed a transcriptome analysis of HGFR182 and compared it with the epithelial R182 cells. There were 2375 DEGs when HGFR182 was compared to R182 cells (Figure 9A). Pathway analysis showed 60 differentially regulated biological pathways (Figure 9B) with seven pathways being related to the EMT process including the TGF-β signaling pathway, focal adhesion, hippo signaling, axon guidance, ECM–receptor interactions, gap junction, and cell adhesion molecules (Figure 9C). 

GO enrichment analysis showed five differentially regulated biological processes between HGFR182 and R182. Interestingly, all of these biological processes were related to EMT, namely ECM organization, positive regulation of Erk1 and Erk2 cascade, positive regulation of cell migration, heterophilic cell–cell adhesion via plasma membrane cell adhesion molecules, and cell adhesion (Figure 9D). These results demonstrate that HGF is able to establish an intermedial stage by inducing the expression of genes necessary to acquire anoikis resistance, cell migration, and potential tumor formation capacity.

## 4. Discussion

In this current study, we report on the molecular characteristics and properties of the E/M hybrid state in ovarian cancer. We demonstrated that the E/M hybrid state represents a stable state, with anoikis resistance and invasive capacity. Furthermore, we identified the HGF/c-MET axis as the main signaling pathway that induces the EOCCs to enter the E/M hybrid state. 

The first interesting finding of this study was the realization that very few ovarian cancer cell lines maintain a full epithelial phenotype, and that there is a wide range of variation in terms of the capacity of these cells to survive detachment, invade, and form tumors. A similar observation was reported by Huang et al. [25]. Clearly, malignant cells do not need to reach the full mesenchymal phenotype in order to be able to resist anoikis and acquire invasive and tumorigenic capacities. In other words, cancer cells with epithelial molecular characteristics may already have acquired metastatic potential and will be responsive to additional environmental signals in order to detach and migrate.

We tested this hypothesis using an in vitro model that recapitulates the acquisition of mesenchymal characteristics while maintaining the morphology and the expression of epithelial genes. Using this model, we identified an early stage, which maintains some epithelial characteristics, but also expresses some early mesenchymal genes. Pathway analysis based on DEGs comparing the E/M hybrid and epithelial stage revealed an upregulation in genes associated with ECM remodeling, TGF-β signaling, and focal adhesion. Furthermore, some of these genes were not activated in the phase associated with the acquisition of the full mesenchymal characteristics. 

Our data showed that genes such as MMP-3, -1, and TNC are only expressed in the early stages of differentiation and that their function involves cleaving a variety of ECM substrates and its reorganization. MMP3 is one member of the metalloproteinases, grouped as stromelysin, known for its ability to cleave different ECM substrates and other MMPs including collagens (III–V and IX), laminin, MMP-2, -7, -8, -9, and -13 [26]. Increased MMP3 expression has been linked to poor prognosis and shorter overall survival in patients with breast, lung, and pancreatic cancers [27]. Treatment with exogenous MMP-3 has been reported to induce EMT in the lung epithelial cells via cleavage of E-cadherin and resulted in the activation of β-catenin pathway [28]. Similarly, an aberrant expression of MMP-1 has been shown to be associated with poor prognosis in colorectal cancer [29]. MMP-1 is a metalloproteinase classified as collagenase with functions in breaking down ECM substrates including collagens (I–III, VII, VIII, and X), MMP-2, and -9 [26]. The expression of both MMP-3 and MMP-1 has been linked with cancer invasion and metastasis [30]. 

Furthermore, there is a significant increase in TNC expression in the E/M hybrid state. TNC is an ECM protein that has many known binding partners including other matrix components, soluble factors, and pathogens [31]. The high expression of TNC within the tumor microenvironment has been shown to promote migration and metastatic potential in tumor cells. One of the well-studied TNC functions is the modulation of cell adhesion to other ECM proteins. The interaction of TNC with other ECM proteins may exert either pro- or anti-adhesion signals. For instance, direct interaction between TNC and fibronectin (FN) results in anti-adhesion by interfering in FN binding with syndecan-4 [32]. TNC-induced invasion and the loss of cell adhesion has been shown to be critical in the initiation of the EMT process in breast cancer [33]. Additionally, TNC has been shown to have an important role in actin cytoskeleton remodeling by the inhibition of RhoA activation in human colon cancer cells [34]. 

Based on these findings, we established a gene signature that we propose represents the E/M hybrid state, which includes an increased expression in MMPs (such as MMP-3 and MMP-1) and TNC, along with the expression of classical epithelial markers (such as CK18 and b-catenin) and mesenchymal markers (such as SOX2 and Vimentin). The transition from fully epithelial to the E/M hybrid state is regulated by factors originating from the microenvironment. We observed that HGF is one of differentially expressed genes in the E/M hybrid state. HGF, also known as a scatter factor, is a growth factor that was first discovered in hepatocyte cells. HGF has been shown to induce migration and invasion in different cancer types [35,36]. 

We tested the hypothesis of whether the HGF treatment of epithelial cells could recreate the early modifications leading to the E/M hybrid state in our model. Indeed, we found that HGF could induce the expression of genes associated with ECM remodeling including MMP-1, MMP-3, and TNC, which, as above-mentioned, are overexpressed in the E/M hybrid cells. Furthermore, HGF treatment in vitro can rescue epithelial cells from anoikis and support invasion. A previous study reported that TNC and HGF could together provide a convergent pro-invasive signal in human colon cancer cells by inactivating RhoA and activating Rac, respectively [34].

However, a central question in the understanding of this process has been the definition of the cell state with the ability to metastasize and whether this state is dependent on the cues from the microenvironment. In order to assess the stability of the E/M hybrid cells, we cultured the cells that had already gained anoikis resistance from HGF treatment in tissue culture treated without the presence of HGF. These cells have persistent anoikis resistance and invasive capacity. They also maintained the expression of epithelial and mesenchymal markers. Our data therefore suggest that the HGF-induced E/M hybrid state has stable anoikis resistance and invasive capacity. Interestingly, based on the analysis of DEGs when the HGF-induced E/M hybrid state (HGFR182) is compared to its epithelial state (R182), we were able to predict which upregulated genes may result in the cellular phenotype we observed (Figure 9E). These DEGs include genes in the laminin family such as LAMA1, LAMA3, and LAMA4, which are known to have important roles in cellular attachment and migration, and have been shown to be overexpressed in ovarian cancer tissues compared to normal tissues, and their expression was positively correlated with worse overall survival and progression-free survival [37]. The other upregulated genes in HGFR182 compared to R182 known to be related to increased cellular invasion are early growth response 1 (EGR1) [38] and prostaglandin-endoperoxide synthase2 (PTGS2) [39]. Additionally, secreted protein acidic and rich in cysteine (SPARC) is known as a key regulator of apoptosis and invasion in ovarian cancer [40]. Finally, we predict that the HGF-induced tumorigenic potential could be due to the upregulation of the Erb-B2 receptor tyrosine kinase3 (ERBB3), which is known to promote the progression and metastasis in ovarian cancer through the IGF1R/STAT2 signaling axis [41].

## 5. Conclusions

In conclusion, we presented the transcriptome analysis of the E/M hybrid state in ovarian cancer cells, and revealed the stability of the invasive phenotype of these cells. Our data suggest that HGF-induced E/M hybrid cells have long-lasting phenotypes of anoikis resistance, migratory, and invasive potential known to support tumor formation. These findings can be potentially used to develop new therapeutic approaches to inhibit the formation of metastasis.

## Figures and Tables

**Figure 1 cancers-15-00684-f001:**
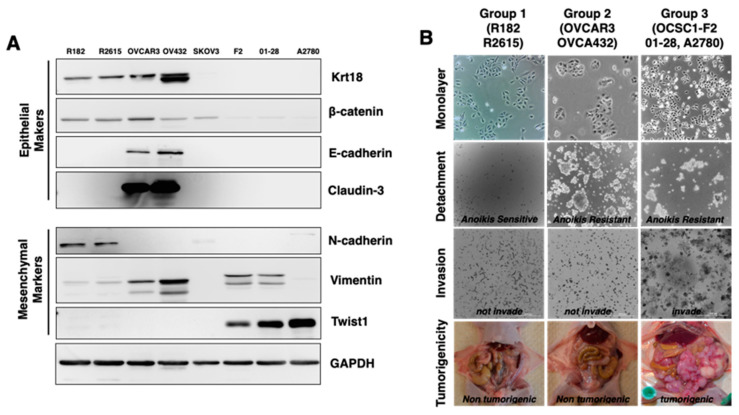
Characterization of the epithelial–mesenchymal molecular and biological phenotypes in ovarian cancer cell lines. (**A**) The EMT molecular characterization was achieved using Western blot with Keratin-18 (Krt-18), β-catenin, E-cadherin, Claudin-3 as the epithelial markers and N-cadherin, Vimentin, and Twist1 as the mesenchymal markers. GAPDH was used as the loading control. (**B**) The EMT biological phenotype characterizations were conducted by anoikis, invasion, and tumorigenicity assays. Cells were grouped into three groups based on their phenotype. The first group (R182 and R2615) was anoikis sensitive, non-invasive, and non-tumorigenic. R182 is shown as representative images. The second group (OVCAR3 and OVCA432) was anoikis resistant, had limited invasion capacity, and tumorigenic potential. OVCAR3 is shown as representative images. The last group (OCSC1-F2, 01-28, and A2780) was anoikis resistant, invasive, and had high tumorigenic potential. OCSC1-F2 is shown as representative images. The images for other cells not shown here are included in Appendix A.

**Figure 2 cancers-15-00684-f002:**
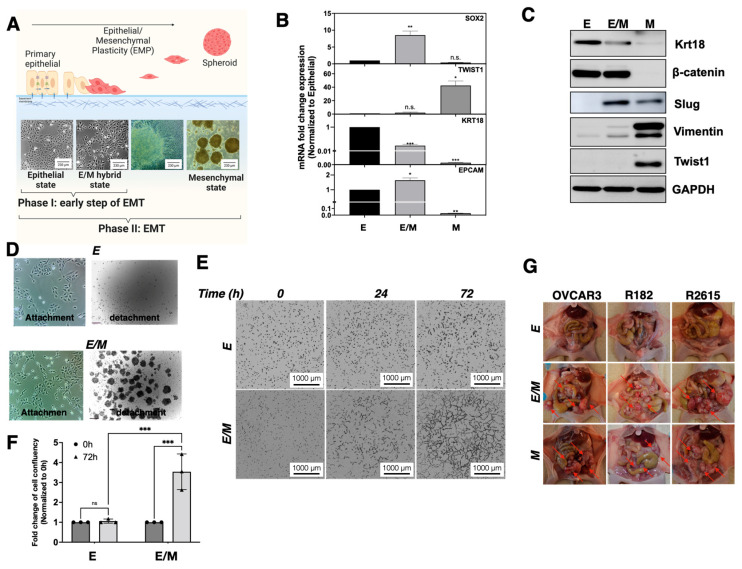
Characterization of the E/M hybrid state in the EMT model in vitro. (**A**) Scheme of the classical EMT differentiation process and representative microscopic images of different phases of EMT progression. The EMT process was completed by 12 days in vitro. The process starts with cells in their epithelial state (day 0); by day 2, we classified the cells as the E/M hybrid state (Phase I of EMT process); by day 7, clusters of differentiated cells were observed; and on day 12, spheroids were observed (referred to as the mesenchymal state) (Phase II of the EMT process). (**B**) Fold change expression of SOX2, TWIST1, KRT18, and EPCAM mRNA as assessed by q-PCR from the E/M hybrid (E/M) and mesenchymal (M) compared to the epithelial (**E**) cells. Bars represent means ± SEM; *p*-value * < 0.05 and ** < 0.01, n.s. = non-significant. One-way ANOVA was used to define the *p*-value. (**C**) Western blot was used to assess the epithelial and mesenchymal markers between E, E/M, and M cells. (**D**) Anoikis assay of E/M cells with anoikis resistance shown by the formation of spheroids whereas E cells are anoikis sensitive. (**E**) Invasion assay comparing E/M and E cells at 0, 24, 72 h following cell seeding in Matrigel (acting as matrix). E/M cell invasion is shown by elongation of cells and growth within the matrix but not with E cells. (**F**) Fold change of cell confluency normalized to E cells. Bars represent means ± SEM; *p*-value *** < 0.001, n.s. = non-significant. Two-way ANOVA was used to define the *p*-value. (**G**) The tumorigenicity assay was conducted by intraperitoneal inoculation of different ovarian cancer cell lines (OVCAR3, R182, and R2615) in different states of EMT in balb/c nude mice. Animals were sacrificed 33 days after inoculation.

**Figure 3 cancers-15-00684-f003:**
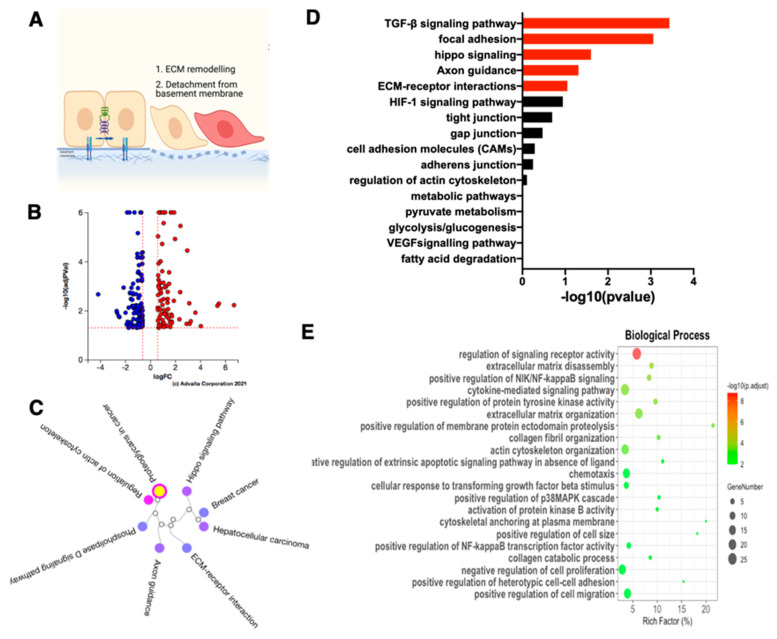
RNA-sequencing analysis of the E/M hybrid from the ovarian cancer cell lines. The E/M hybrid state cells of R182, R2615, and OVCAR3 were compared with their respective Estate cells in this analysis. (**A**) Proposed scheme of the first few steps in E differentiation to the E/M state including ECM remodeling and detachment from the basement membrane. (**B**) Volcano plot of differentially expressed genes (DEGs); blue dots represent downregulated DEGs, red dots represent upregulated DEGs. (**C**) Dendrogram of significant biological pathways in E/M cells compared to E cells. (**D**) Bar plot of the EMT related biological pathways in E/M compared to E cells. Significant pathways are presented as red bars, and non-significant pathways as black bars. (**E**) Top 20 GO terms for biological processes analysis in E/M when compared to the E cells.

**Figure 4 cancers-15-00684-f004:**
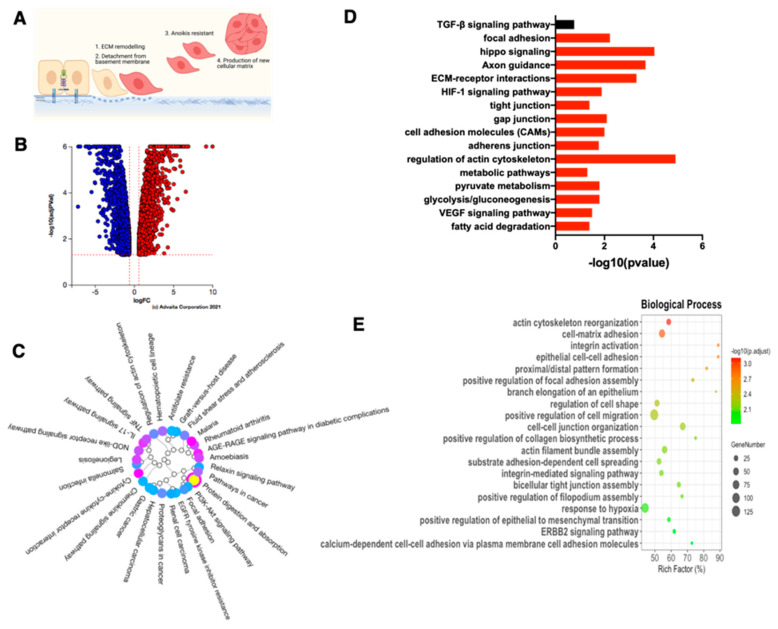
RNA-sequencing analysis of the mesenchymal state of the ovarian cancer cell lines. The M state cells of R182, R2615, and OVCAR3 were compared to their respective E state cells in this analysis. (**A**) Proposed scheme of the completed steps in E differentiation to the M state include (1) ECM remodeling; (2) detachment from the basement membrane; (3) anoikis resistance; and (4) the production of new cellular matrix. (**B**) Volcano plot of differentially expressed genes (DEGs); blue dots represent downregulated DEGs, red dots represent upregulated DEGs. (**C**) Dendrogram of significant biological pathways in M compared to E cells. (**D**) Bar plot of the EMT-related biological pathways in M compared to E cells. Significant pathways are presented as red bars, and non-significant pathways as black bars. (**E**) Top 20 of GO terms for the analysis of the biological processes in M when compared to E cells.

**Figure 5 cancers-15-00684-f005:**
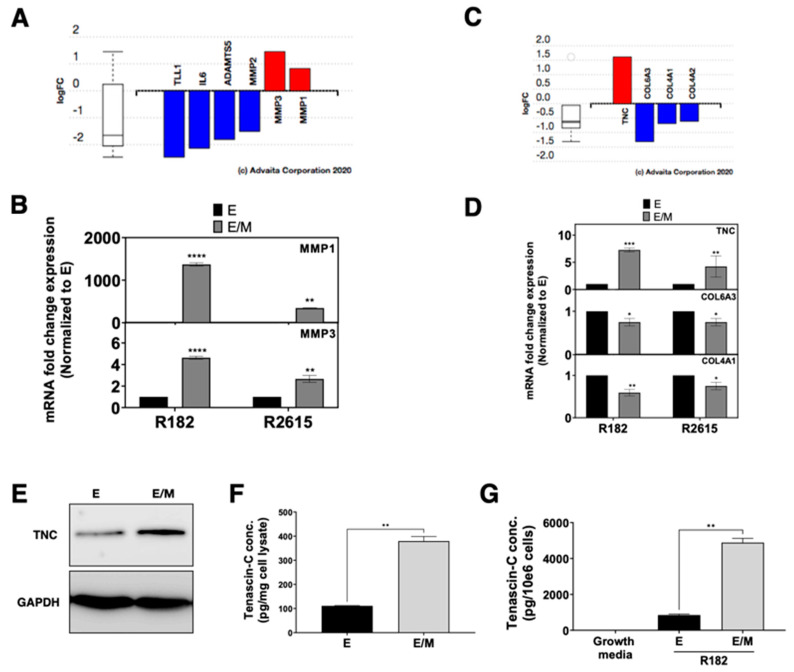
Differentially expressed genes (DEGs) related to ECM disassembly and organization in the E/M hybrid compared to the E cells. (**A**) Bar plot of log2 fold change (logFC) of DEGs related to ECM disassembly. (**B**) Validation of MMP1 and MMP3 mRNA expression quantified by q-PCR in the E/M and M state from the R182 and R2615 cells. Bars represent the means ± SEM; *p*-value ** < 0.01 and **** < 0.0001, n.s. = non-significant. One-way ANOVA was used to define the *p*-value. (**C**) Bar plot of the log FC of DEGs related to ECM organization. (**D**) Validation of Tenascin-C (TNC), Collagen6A3 (COL6A3), and Collagen4A1 (COL4A1) mRNA expression quantified by q-PCR in the E/M and M state from the R182 and R2615 cells. Bars represent the means ± SEM; *p*-value * < 0.05, ** < 0.01 and *** < 0.001, n.s. = non-significant. One-way ANOVA was used to define the *p*-value. (**E**) Cellular TNC protein expression in E and E/M state of R182. Quantification of TNC using ELISA for (**F**) intracellular and (**G**) extracellular (secreted) expression. Bars represent means ± SEM; *p*-value ** < 0.01. Student *t*-test was used to define the *p*-value.

**Figure 6 cancers-15-00684-f006:**
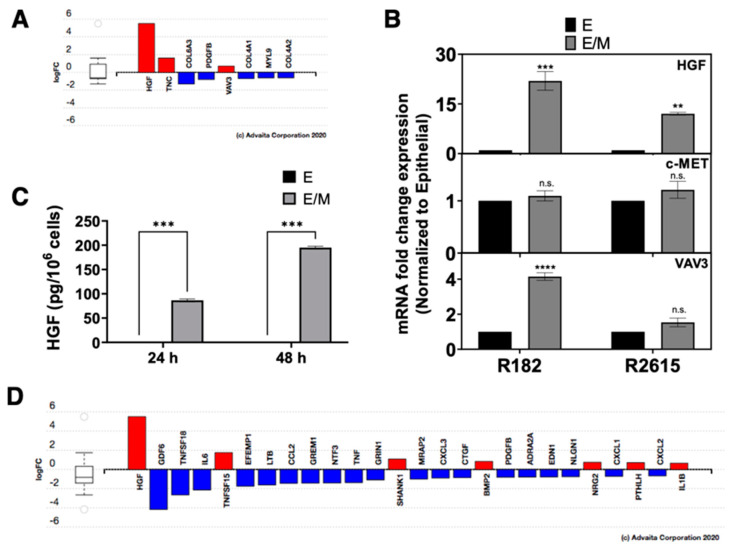
DEGs related to focal adhesion reorganization and signaling receptor activity in the E/M hybrid compared to E cells. (**A**) Bar plot of the logFC of DEGs related to focal adhesion reorganization. (**B**) Validation of HGF, c-MET, and VAV3 mRNA expression quantified by q-PCR in the E/M and M state from the R182 and R2615 cells. Bars represent the means ± SEM; *p*-value ** < 0.01 and *** < 0.001, n.s. = non-significant. One-way ANOVA was used to define the *p*-value. (**C**) Quantification of secreted HGF using the ELISA-based assay for the E and E/M cells for 24 and 48 h of culture in OPTIMEM media. Bars represent the means ± SEM; *p*-value *** < 0.001. Student *t*-test was used to define the *p*-value. (**D**) Bar plot of the log FC of the DEGs related to signaling receptor activity. **** < 0.0001.

**Figure 7 cancers-15-00684-f007:**
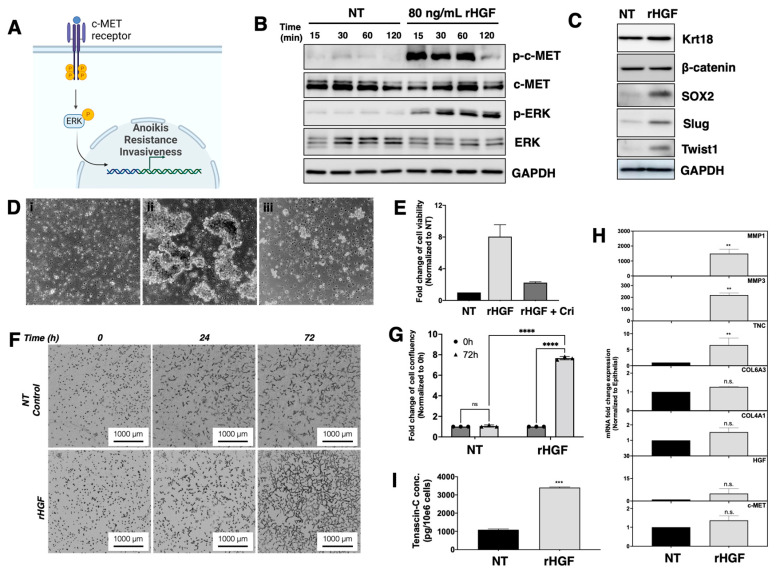
The effect of HGF treatment in the epithelial ovarian cancer cells. (**A**) Proposed scheme of the HGF pathway in ovarian cancer cells. (**B**) Characterization of the classical HGF pathway was achieved by the Western blot of R182 treated with 80 ng/mL rHGF for 15–120 min compared to their respective NT control for their expression of phosphorylated c-MET (p-c-MET), total c-MET, phosphorylated ERK, and total ERK. (**C**) Characterization of EMT markers following 80 ng/mL rHGF treatment in R182. (**D**) Anoikis assay for R182: (i) NT control, (ii) 80 ng/mL rHGF, and (iii) 80 ng/mL rHGF and 100 nM Crizotinib. (**E**) Quantification of cell viability for the anoikis assay presented as fold change normalized to NT control. (**F**) Invasion assay of R182 in the presence or absence of 80 ng/mL rHGF for 0, 24, and 72 h. (**G**) Quantification of the invasion assay presented as a fold change of cell confluency normalized to cells in 0 h (time of seeding). Bars represent means ± SEM; *p*-value **** < 0.0001, n.s. = non-significant. Two-way ANOVA was used to define the *p*-value. (**H**) Bar plot of qPCR for MMP1, MMP3, TNC, COL6A3, COL4A1, HGF, and c-MET mRNA expression. GAPDH expression was used as the house-keeping gene. Bars represent the means ± SEM; *p*-value ** < 0.01, n.s. = non-significant. Student *t*-test was used to define *p*-value. (**I**) Quantification of secreted TNC concentration following 80 ng/mL rHGF treatment for 48 h. Bars represent the means ± SEM; *p*-value *** < 0.001. Student *t*-test was used to define the *p*-value.

**Figure 8 cancers-15-00684-f008:**
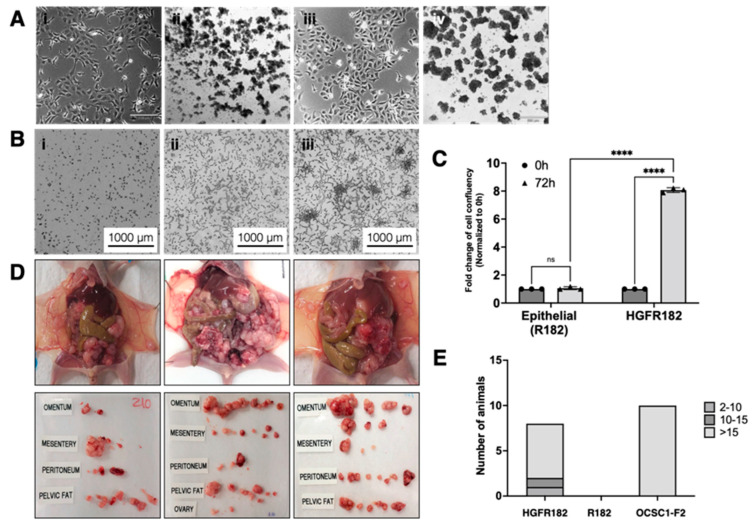
Generation of the HGF-treated E/M hybrid cells and their characterization. (**A**) Cellular images of (i) monolayer R182 cells post 80 ng/mL rHGF treatment for 48 h, (ii) R182 cells post 80 ng/mL rHGF treatment for 48 h seeded in ULA plate with the addition of 80 ng/mL rHGF cultured for 7 days, (iii) cells from condition (ii) plated back in the tissue culture-treated surface as a monolayer (referred as HGFR182), (iv) cells from the condition shown in (iii) cultured in the ULA plate without the addition of rHGF for 7 days. (**B**) Invasion assay of HGFR182 (i) 0, (ii) 24, and (iii) 72 h timepoints. (**C**) Quantification of the invasion assay presented as the fold change of cell confluency normalized to cells in 0 h (time of seeding). Bars represent the means ± SEM; *p*-value **** < 0.0001, n.s. = non-significant. Two-way ANOVA was used to define the *p*-value. (**D**) Tumorigenicity assay of the HGFR182 cells shown in the three representative animals 33 days following the inoculation of HGFR182 cells and three images of the collected macroscopic tumor implants from the representative three animals. (**E**) Bar plot of the number of animals with either 2–10, 10–15, and >15 tumor implants for animals inoculated with HGFR182, R182, and OSCS1-F2.

**Figure 9 cancers-15-00684-f009:**
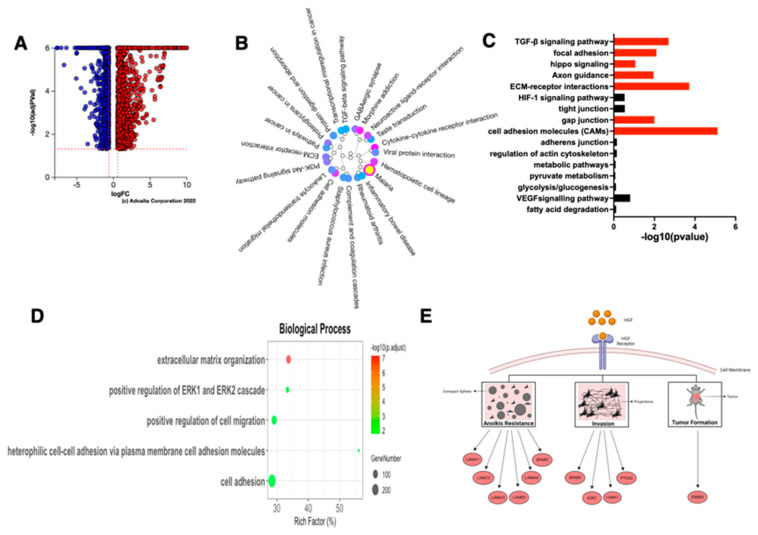
RNA-sequencing analysis of the HGF-treated E/M hybrid (HGFR182) compared to its epithelial counterpart (R182). (**A**) Volcano plot of DEGs; blue dots represent downregulated DEGs, red dots represent upregulated DEGs. (**B**) Dendrogram of significant biological pathways in the HGFR182 compared to R182 cells. (**C**) Bar plot of EMT related biological pathways in HGFR182 compared to R182 cells. Significant pathways are presented as red bars, and non-significant pathways as black bars. (**D**) GO terms for the analysis of biological processes in HGFR182 when compared to R182 cells. (**E**) Predicted scheme of the cellular phenotype observed with the analyzed DEGs when HGFR182 was compared to R182.

## Data Availability

The data presented in this study are openly available in GEO at accession number GSE22329.

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
