# Peer review of "Generation of Stable Epithelial–Mesenchymal Hybrid Cancer Cells with Tumorigenic Potential"

_cancers, 2023, doi:10.3390/cancers15030684_

Round 1

Reviewer 1 Report

The intro and abstract part presented epithelial mesenchymal hybrid cells in  cancer; however, it must have been more specifically stated in ovarian cancer. 

The references are not uniform format, and more than 15% self-citations.

Author Response

must have been more specifically stated in ovarian cancer. 

Response: we have added a paragraph about ovarian cancer and metastasis

The references are not uniform format, and more than 15% self-citations.

Response: We have added relevant references

L-17- epithelial and mesenchymal like properties

Response: “different degree of epithelial and mesenchymal” has been replaced with “epithelial and mesenchymal like properties”

L-18- Characteristics shouldn’t be capital C

Response: manuscript has been modified accordingly

L-21- the different steps during the transition of cancer cells into E/M hybrid state.

Response: “the different steps within the differentiation process resulting on E/M hybrid state.” has been modified to “the different steps during the transition of cancer cells into E/M hybrid state.”

L -92- put the ref (16)

Response: manuscript has been modified accordingly

For Fig 1-B- Please use Group I, Group 2, and Group 3 and in parentheses use the representative actual cell line names on the top of each group. Using cell lines names at the top just making the figure confusing.

Response: figure has been modified accordingly

L-181-ULA abr needs to be defined "ultra-low adherent" -because in the materials and methods section, the protocol is only ref as previous paper.

Response: manuscript has been modified accordingly

L-197- 5x10^6

Response: manuscript has been modified accordingly

L-211- How about considering the heterogeneity of cells within a cell line. Have you try FLOW CYTOMETRIC analysis of these cell lines and if you can see a subpopulation to sort them out.!!

Respond: thank you for the suggestions, indeed, in previous studies, we performed Flow cytometry and showed that the used cells lines are homogeneous.  (Alvero AB, Chen R, Fu HH, Montagna M, Schwartz PE, Rutherford T, Silasi DA, Steffensen KD, Waldstrom M, Visintin I, Mor G. Molecular phenotyping of human ovarian cancer stem cells unravels the mechanisms for repair and chemoresistance. Cell Cycle. 2009 Jan 1;8(1):158-66. doi: 10.4161/cc.8.1.7533. PMID: 19158483; PMCID: PMC3041590.)

L-218- put [] for ref 20-23

Response: manuscript has been modified accordingly

Reviewer 2 Report

The authors present a comprehensive study on EMT in breast cancer cell lines including transcriptome and functional analysis. Praticularly, they pinpoint, that anoikis resitance, invasive capacity and tumorigenic potential is stably aquired already in E/M hybrid states. The identification of HGF/c-Met pathway as a drive of EMT is a very interesting conclusion.

Minor comments:

Some figure panels are difficult to read (e.g. Fig 2A,F; Fig 3B, Fig 4A,B; Fig 7A,E,G-I and others)

Some typing errors need correction:

Line 92: "publication16"

Line 197: "5 106"

The letter G in group is sometimes capitalised and sometimes lower case throughout the manuscript.

Author Response

The authors present a comprehensive study on EMT in breast cancer cell lines including transcriptome and functional analysis. Particularly, they pinpoint, that anoikis resitance, invasive capacity and tumorigenic potential is stably acquired already in E/M hybrid states. The identification of HGF/c-Met pathway as a drive of EMT is a very interesting conclusion.

Minor comments:

Some figure panels are difficult to read (e.g. Fig 2A,F; Fig 3B, Fig 4A,B; Fig 7A,E,G-I and others)

Response: manuscript has been modified accordingly

Some typing errors need correction:

Line 92: "publication16"

Response: manuscript has been modified accordingly

Line 197: "5 106"

Response: manuscript has been modified accordingly

The letter G in group is sometimes capitalised and sometimes lower case throughout the manuscript.

Response: manuscript has been modified accordingly